# Inter-Site Cooperativity of Calmodulin N-Terminal Domain and Phosphorylation Synergistically Improve the Affinity and Selectivity for Uranyl

**DOI:** 10.3390/biom12111703

**Published:** 2022-11-17

**Authors:** Maria Rosa Beccia, Sandrine Sauge-Merle, Nicolas Brémond, David Lemaire, Pierre Henri, Christine Battesti, Philippe Guilbaud, Serge Crouzy, Catherine Berthomieu

**Affiliations:** 1CEA, CNRS, UMR 7265, BIAM, Interactions Protéine Métal, Aix-Marseille University, 13108 Saint-Paul-lez-Durance, France; 2LPC2E, CNRS, University Orléans, 45071 Orléans, France; 3Laboratoire Lagrange, Observatoire Côte d’Azur, Université Côte d’Azur, CNRS, CEDEX 4, 06304 Nice, France; 4CEA, DES, ISEC, DMRC, Département de Recherche sur les Procédés pour la Mine et le Recyclage du Combustible, University Montpellier, Marcoule, France, 30207 Bagnols-sur-Cèze, France; 5Groupe de Modélisation et Chimie Théorique, IRIG, UMR CEA, CNRS, Université Joseph Fourier, CEDEX 9, 38054 Grenoble, France

**Keywords:** uranium, protein, cooperativity, metal binding competition, thermodynamics, spectroscopy

## Abstract

Uranyl–protein interactions participate in uranyl trafficking or toxicity to cells. In addition to their qualitative identification, thermodynamic data are needed to predict predominant mechanisms that they mediate in vivo. We previously showed that uranyl can substitute calcium at the canonical EF-hand binding motif of calmodulin (CaM) site I. Here, we investigate thermodynamic properties of uranyl interaction with site II and with the whole CaM N-terminal domain by spectrofluorimetry and ITC. Site II has an affinity for uranyl about 10 times lower than site I. Uranyl binding at site I is exothermic with a large enthalpic contribution, while for site II, the enthalpic contribution to the Gibbs free energy of binding is about 10 times lower than the entropic term. For the N–terminal domain, macroscopic binding constants for uranyl are two to three orders of magnitude higher than for calcium. A positive cooperative process driven by entropy increases the second uranyl-binding event as compared with the first one, with ΔΔG = −2.0 ± 0.4 kJ mol^−1^, vs. ΔΔG = −6.1 ± 0.1 kJ mol^−1^ for calcium. Site I phosphorylation largely increases both site I and site II affinity for uranyl and uranyl-binding cooperativity. Combining site I phosphorylation and site II Thr7Trp mutation leads to picomolar dissociation constants Kd_1_ = 1.7 ± 0.3 pM and Kd_2_ = 196 ± 21 pM at pH 7. A structural model obtained by MD simulations suggests a structural role of site I phosphorylation in the affinity modulation.

## 1. Introduction

Uranium is a radioactive metal, naturally present in the Earth’s crust, which is exploited for the nuclear industry and for military applications. The risk of its distribution in the environment is associated with mining activities, uranium processing, leaching of radioactive wastes or accidental situations in nuclear power plants. Uranium presents radiological and chemical toxicity to living organisms [1,2,3] and its release into the environment can constitute a serious hazard for ecosystems and human health. There is increasing evidence that uranium can substitute biological metals, such as iron or calcium, in essential metalloproteins such as transferrin, osteopontin, fetuin, C-reactive protein or calmodulin, as examples, in [4,5,6,7,8,9,10,11,12,13]. Transferrin is the major iron carrier in blood, osteopontin and fetuin are involved in calcium metabolism and associated mineralization processes, and calmodulin is a calcium-dependent signaling protein present in all eukaryotic cells, which modulates more than 100 target proteins involved in numerous physiological processes including cell proliferation or programmed cell death [14,15]. For the active site of C-reactive protein or for calmodulin metal-binding site I (see below), uranium-binding affinity was shown to be significantly greater than that of calcium [8,16]. In addition, uranium binding to the C-reactive protein was shown to impair interaction with its cognate substrate phosphorylcholine [8] and recently, uranium binding to calmodulin was shown to decrease the calmodulin-dependent enzymatic activity of phosphodiesterase, highlighting the potential of uranium to impair the function of calmodulin as a signaling molecule [17]. These data illustrate that the chemical toxicity of uranium may be associated with its capacity to compete with biological cations, bind to cellular constituents as proteins and impair their physiological role. It is thus of prime interest to better understand structural factors governing uranium thermodynamic stabilization in proteins to unravel and predict its chemical toxicity in vivo.

As hard Lewis acids, uranium and calcium form electrostatic interactions preferentially with hard donor oxygen ligands, provided by carbonyl, carboxylate, phenolate, hydroxyl or phosphoryl groups in proteins [18,19]. In addition, uranium is present as the dioxo uranyl UO_2_^2+^ cation in biological media, and its coordination geometry with five to six ligands in the equatorial plane is similar to the Ca^2+^ coordination geometry of seven ligands arranged in pentagonal bipyramidal structures [18].

Calmodulin (CaM) is a prototypical example of the family of EF-hand calcium-binding proteins functioning as Ca^2+^- buffer or Ca^2+^- sensors in cells [20,21]. CaM is composed of two globular N-terminal (N-ter) and C-terminal domains, each containing a pair of EF-hand Ca(II)-binding motifs, denoted site I and site II for the N-ter domain. In these helix-loop-helix motifs, the Ca(II) ligands are located at positions 1, 3, 5, 7, 9 and 12 of the loop (Figure 1) [22,23,24,25,26]. 

The CaM N-ter domain is an interesting platform for the engineering of metal-binding sites and for addressing different factors that control uranium-binding affinity and selectivity in proteins. It has a structured conformation both in the apo- and in the metal-bound forms, all metal ligands are gathered on short binding loops, and the metal-binding process at sites I and II can be cooperative [27,28]. It is also an interesting scaffold to develop short peptides for the selective and affine binding of metal cations [29,30,31,32].

We previously showed that the uranyl complex formed by the CaM site I is more stable than that formed with calcium, with conditional K_d_ values of 25 ± 6 nM and 38.4 ± 1.0 µM, respectively, at pH 6 (logK = 7.6 and 4.6, respectively) [16]. In addition, phosphorylation of the threonine at position 9 (Thr9_P_) in site I significantly increased the uranyl-binding affinity, with K_d_ = 5 ± 1 nM at pH 6 and K_d_ = 320 ± 57 pM at pH 7 (logK = 8.3 and 9.5, respectively). FTIR and EXAFS data showed the direct interaction of the deprotonated phosphoryl group of Thr9_P_ with uranyl at pH 7 [16,33]. Increased affinity for uranyl could thus be associated with the introduction of a strong phosphoryl ligand in the uranyl coordination sphere. Substitution of glutamate by phosphoserine in cyclic peptides engineered with a preorganized backbone β-sheet structure also significantly increased the uranium-binding affinity, by one order of magnitude from logK = 8.2 to 9.2 for the monosubstituted peptides, and up to logK = 11.3 for the tetraphosphorylated peptide [34,35,36,37].

We also showed that the structure of the metal-binding loop in CaM site I has a strong effect on uranyl-binding affinity. An increase in affinity by two orders of magnitude from logK = 7.6 to logK = 9.7 was obtained by suppressing two amino acids of the metal-binding loop, shortening this loop from 12 to 10 amino acids [38]. These results also suggested that the addition of phosphoryl groups must be associated with structural considerations to optimize very affine and specific binding sites.

In this regard, allostery is an interesting property of proteins that increases the protein sensitivity to small changes in ligand concentration. In CaM, Ca(II) binding is governed by intra- and inter- domain cooperative effects [27,39,40]. We previously obtained a quantitative view of the cooperativity in the CaM N-ter domain by comparing the Ca(II) binding properties of the CaM N-ter domain with functional sites I and II (referred to as two-site peptides) and with site I or site II impaired for Ca(II) fixation (one-site peptides) [28]. We found that site I affinity for Ca(II) was ≈ 1.5 times that of site II and that cooperativity induced an approximately tenfold higher affinity for the second Ca(II)-binding event as compared to the first one. We further showed that insertion of a tryptophan at position seven of the site II binding loop significantly increased site II’s affinity for Ca(II) and the intra-domain cooperativity [28]. 

In the present study, we investigate the thermodynamic properties of uranyl interaction with site II and the CaM N-ter domain by spectrofluorimetry and calorimetry (ITC), compare them with those of calcium, and assess uranium-binding cooperativity. We analyze one-site and two-site peptides with native sequences, sequences modified by the introduction of a tryptophan at position seven of site II binding loop, and two-site peptides with a phosphothreonine at position nine of site I (Table 1). 

We show that CaM site II has an affinity for uranyl about 10 times lower than that of site I, and that the behavior of uranyl binding to the CaM N-ter domain can be rationalized by a small positive uranyl-binding cooperativity. Interestingly, phosphorylation of Thr9 at site I significantly enhances the uranyl-binding affinity at site II and very high affinities for uranyl are observed for the phosphorylated two-site peptides, with macroscopic dissociation constants in the picomolar to subnanomolar range at pH 7. A structural model obtained by molecular dynamics (MD) simulations illustrates how the structure of the phosphorylated two-site N-ter domain can influence both site I and site II’s affinity for uranyl.

## 2. Experimental Section

### 2.1. Engineering, Expression and Purification of CaM Peptides

The CaM-D1 construct containing the *Arabidopsis thaliana* sequence of the CaM N-ter domain was obtained as previously described [16] and used as a template for new mutant constructs. All point mutations were produced with the QuickChange site-directed mutagenesis kit (Stratagene) and specific primer pairs (Appendix A), according to the manufacturer’s instructions. Each mutated DNA sample was amplified with the D1-CaM-TEV-S and D1-CaM-STOP-AS primers to introduce the TEV protease recognition site upstream of the coding sequence as described in [16]. Protein expression and purification were performed as previously described [16].

### 2.2. In Vitro Phosphorylation of the CaM Peptides

In vitro phosphorylation assays for CaM1 Y I-II and CaM1 YW I-II peptides were performed in vitro using the recombinant alpha subunit of casein kinase II (αCK2) as described in [16,28].

### 2.3. Chemicals and Stock Solutions

MES buffer solution (20 mM, pH 6) and Tris buffer solution (20 mM, pH 7) were prepared by weighting the appropriate amount of the analytical grade salt (99.5%, Sigma-Aldrich, Saint Louis, MI, USA) and dissolving it in pure water. Solid KCl was added to each buffer solution to yield a final concentration of 100 mM. The uranyl solutions were prepared by diluting a 0.1 M stock solution of uranyl nitrate (pH 3.5, stored frozen at −20 °C) in the final buffer. Calcium chloride solution was prepared by dissolving the appropriate amount of analytical grade CaCl_2_·2H_2_O (99.5%, Merck) in pure water. All solutions were prepared with ultrapure water (18 MΩ). The pH values of the solutions were measured with a pH electrode (IoLine, Schott Instruments, Mainz, Germany) and pH meter (Mettler Toledo, Greifensee, Switzerland); the electrode was calibrated with standard buffers. The pH values of buffer solutions were adjusted with NaOH (10 N, Sigma-Aldrich, Saint Louis, MI, USA) or HCl (37%, Sigma-Aldrich, Saint Louis, MI, USA). Before use, buffer solutions and protein solutions were treated to remove any trace of calcium as previously described [28]. 

### 2.4. Mass Spectrometry Analyses

Mass spectrometry (MS) analyses were performed on a MicroTOF-Q (Bruker Daltonik GmbH, Bruker Billerica, MA, USA) with an electrospray ionization source (ESI) as previously described [16,28]. Data were acquired in the positive mode and calibration was performed using a calibrating solution of ESI Tune Mix (Agilent Technologies, Santa Clara, CA, USA) in CH_3_CN/H_2_O (95/5 *v*/*v*). The system was controlled by the software package MicroTOF Control 2.2- and data were processed with DataAnalysis 3.4 both provided by Bruker Daltonik GmbH.

### 2.5. Spectrofluorimetric Titrations

The binding affinity of CaM mutants for uranyl ion was measured by monitoring the fluorescence intensity of a single aromatic residue: Tyr7 of site I (302 nm) for the “CaM Y” labelled peptides or Trp7 of site II (350 nm) for the “CaM YW” labelled peptides. Spectrofluorimetric titrations were performed at pH 6 or pH 7 and at an ionic strength I = 0.12 M. For each titration, a peptide solution (10 µM) was prepared in the appropriate buffer solution containing iminodiacetate (IDA, 100 µM). Spectra were acquired both in the absence of and after stepwise additions of increasing UO_2_(NO_3_)_2_ amounts. Spectra were collected on a Cary Eclipse spectrofluorometer (Agilent Technologies, Santa Clara, CA, USA) at 25 °C, with excitation at 270 nm for tyrosine and 260 nm for tryptophan. Emission spectra were recorded from 290 to 400 nm. The excitation and emission slits were 10 nm. 

### 2.6. ITC Titrations

Isothermal titration calorimetry experiments were performed at 298 K using a MicroCal iTC200 device (Cytiva Europe GmbH, Vélizy-Villacoublay, France). The instrument consists of two identical cells, one for the sample and one for the reference solution. The reference cell of the microcalorimeter was filled with ultrapure water, and both cells were maintained at the same temperature. A 200 µL quantity of a solution containing UO_2_(NO_3_)_2_ and IDA (IDA/ UO_2_^2+^ ratio = 2) was deposited in the sample cell during a typical titration, with stirring at 1000 rpm; 2 µL of the protein solution was injected at equal 150 s intervals (0.4 µL on the first injection). In total, 38.4 µL of protein solution was added to the uranyl nitrate-IDA solution. The heat of reaction was calculated at each injection as the difference in heat necessary to maintain the sample cell and reference cell at the same temperature. A blank experiment (without UO_2_(NO_3_)_2_ in the sample cell) was performed for each peptide, in order to subtract the heat effects due to dilution and mixing. For consistency with the fluorescence experiments, ITC measurements were performed at pH 6 (MES buffer, KCl). 

### 2.7. Calculations

#### 2.7.1. Competition with IDA

Iminodiacetate, IDA, was added to peptide solutions (spectrofluorimetric titrations) or to uranyl solutions (ITC titrations) to control uranyl speciation and to avoid the formation of hydroxo-uranyl complexes, which are formed at a pH higher than 4, as previously described [16]. IDA chelates uranyl, forming three different complexes, for whom the stability constants at 25 °C and I = 0.1 M are known [41]. 

#### 2.7.2. Spectrofluorimetric Data Treatment and Evaluation of Macroscopic Constants

A one-site peptide (P), able to bind only one equivalent of the uranyl ion (M), forms a mononuclear complex (MP) according to the following equation:
K_1_
 M + P ⇄ MP(1)

The fluorescence binding isotherms can be fitted to a model described by the following equation:ΔF/C_P_ = ΔΦ K_1_ [M]/(1 + K_1_ [M])(2)
where ΔF is the change in fluorescence signal during titration, C_P_ is the total protein concentration, ΔΦ is the amplitude of the binding isotherm, K_1_ is the conditional equilibrium constant of reaction and [M] is the free metal concentration [28].

For two-site peptides, a dinuclear complex M_2_P is formed through the two-step process described by both Reactions (1) and (3):
K_2_
 M + MP ⇄ M_2_P(3)

The conditional equilibrium constants, K_1_ and K_2_, are linked to the fluorescence signal and the free uranyl concentration [M], according to Equation (4) [28]:ΔF/C_P_ = (ΔΦ_1_ K_1_ [M] + ΔΦ_2_ K_1_ K_2_ [M]^2^)/(1 + K_1_ [M] + K_1_ K_2_ [M]^2^)(4)

In the presence of multiple competing reactions involving M (formation of both uranyl–IDA complexes and uranyl–peptide complexes), the analytical estimation of the free M concentration and, consequently, of K_1_ and K_2_ is not possible. Therefore, [M], K_1_ and K_2_ were calculated for each titration with a numerical Newton–Raphson method, by taking in account all the complex-formation equilibria, the acid-dissociation constants of IDA and the fluorescence variation (Equation (2) or Equation (4), depending on the peptide), as detailed in Supporting information.

#### 2.7.3. Microscopic Constants and Cooperativity Evaluation

To evaluate the peptide inter-site cooperativity, it is necessary to characterize the apo- and holo-peptide, as well as its intermediate microstates. In a two-site peptide, four microstates are possible with the population of each depending on the microscopic thermodynamic constants of the system (Figure 1C). Microscopic constants (K_I_ and K_II_), are related to macroscopic ones (K_1_ and K_2_) according to equations:K_1_ = K_I_ + K_II_(5)
K_2_ = K_I_ K_II,I_/(K_I_ + K_II_)(6)

The inter-site cooperativity was quantified using the parameter ΔΔG, that is, the difference in free energy between uranyl binding to the apo-peptide and uranyl binding to the peptide already loaded with one uranyl ion, as described in [27]. ΔΔG allows to distinguish between a positive (ΔΔG < 0) and a negative cooperativity (ΔΔG > 0). It is related to the microscopic constants, according to the following equation:ΔΔG = −RTln(K_I,II_/K_I_) = −RTln(K_II,I_/K_II_)(7)

#### 2.7.4. Isothermal Titration Calorimetry Data Treatment and Microscopic Thermodynamic Parameters

For ITC measurements, we used the same equilibria described for the fluorescence experiments (Equations (1)–(3)). For the one-site peptides, experimental data were fitted to Equation (8):(8)$$Q=\sum_{i}{\Delta H}_{i}{\Delta n}_{i}$$
where ∆n_i_ is the change in the number of moles of each species in solution, and ∆H_i_ is the respective molar binding enthalpy.

For a cooperative binding in a two-site macromolecule, the heat released or adsorbed at each injection is defined by the equation:Q = ΔH_I_ Δn_MP,I_ + ΔH_II_ ∆n_MP,II_ + (ΔH_I_ + ΔH_II_ + ΔH_c_) ∆n_M2P_(9)
where ΔH_I_ and ΔH_II_ are the molar enthalpies associated with the binding of the metal ion at site I and site II of the apo-protein, respectively; ∆n_MP,I_ and ∆n_MP,II_ correspond to the change in number of moles of the two possible mononuclear complexes (MP_I_, with the metal bound at site I, and MP_II_, with the metal bound at site II) at each injection; ∆n_M2P_ is the change in number of moles of the dinuclear complex, M_2_P, at each injection; ΔH_c_ is the molar enthalpy associated with the cooperative effect [42,43]. ΔH_I_, ΔH_II_ and ΔH_c_ are microscopic parameters. The change in the number of moles, ∆n, between the injection j and the injection j − 1, is related to the concentration of the corresponding species and, consequently, to the equilibrium constants K_I_, K_II_ or K_2_: ∆n_MP,I_ = V_j_[MP_I_]_j_ − V_j_[MP_I_]_j-1_ = (V_j_[M]_j_ [P]_j_ − V_j-1_[M]_j-1_ [P]_j-1_)K_I_(10)
∆n_MP,II_ = V_j_[MP_II_]_j_ − V_j_[MP_II_]_j-1_ = (V_j_[M]_j_ [P]_j_ − V_j-1_[M]_j-1_ [P]_j-1_)K_II_(11)
∆n_M2P_ = V_j_[M_2_P]_j_ − V_j_[M_2_P]_j-1_ = (V_j_[M]_j_ [MP]_j_ − V_j-1_[M]_j-1_ [MP]_j-1_)K_2_(12)

Using Equations (9)–(12), it is possible to evaluate the heat of reaction as a function of the microscopic constants K_I_ and K_II_ and the macroscopic constant K_2_:
Q = ΔH_I_ (V_j_[M]_j_ [P]_j_ − V_j-1_[M]_j-1_ [P]_j-1_)K_I_ + ΔH_II_ (V_j_[M]_j_ [P]_j_ − V_j-1_[M]_j-1_ [P]_j-1_)K_II_ +  (ΔH_I_ + ΔH_II_ + ΔH_c_)(V_j_[M]_j_ [MP]_j_ − V_j-1_[M]_j-1_ [MP]_j-1_)K_2_(13)

Concentrations of M, P and MP were evaluated using the Newton–Raphson method. K_II_ and ΔH_II_ were evaluated from the ITC curves of the one-site peptides, K_I_, ΔH_I_ and K_2_ were obtained as parameters of the fit for the ITC curves of the two-site peptides. The macroscopic constant K_1_ was evaluated from the obtained K_I_ and K_II_, according to Equation (5).

The entropies for the binding reaction were calculated from measured values of K and ΔH of the corresponding process, by means of Equation (14).
ΔG = ΔH − TΔS = –RTlnK(14)

For the two-site peptides, we evaluated the entropy of uranyl binding to site I of the apo-protein (TΔS_I_ = RTlnK_I_ + ΔH_I_), the entropy of uranyl binding to site II of the apo-protein (TΔS_II_ = RTlnK_II_ + ΔH_II_) and the entropy associated with the inter-site cooperativity (TΔS_c_ = RTlnK_c_ + ΔH_c_). K_c_, the cooperativity constant, is an equilibrium constant calculated as K_I,II_/K_I_ = K_II,I_/K_II_ [42].

For both spectrofluorimetric and calorimetric titrations, the reported thermodynamic parameters are averages of three experimental values.

### 2.8. Molecular Dynamics

The initial model was built from the three-dimensional X-ray structure of *Paramecium tetraurelia* calmodulin (pdb code 1N0Y). The structure of CaM1 Y I-II P was completed and refined with the molecular dynamics program CHARMM [44]. The all-atom force field *all36* for proteins [45] was used. All histidines were protonated on their N*δ* atom, aspartate and glutamate residues were kept unprotonated (negative charge) and all arginine and lysine residues were positively charged. Missing hydrogen atom coordinates were built with CHARMM and the structure was energy minimized down to a gradient of 0.1 kcal mol^−1^ Å^−1^ with the ABNR algorithm and subject to harmonic restraints on heavy atoms with force constant of 5 kcal mol^−1^ Å^−2^. Ten sodium counterions were added to neutralize the system. A 66 × 48 × 48 Å^3^ box of TIP3P water (5088 molecules), sufficient to accommodate the protein maintaining all atoms 10 Å away from the boundaries, was energy minimized and equilibrated. The CaM structure was then immersed into that water box (total of 15,157 atoms), energy minimized allowing the box size to vary, and pre-equilibrated with 10 ps Langevin dynamics at a temperature of 300 K with restraints on all backbone atoms of the protein to yield the structure “*NonOe*”. The initial distances between site I Thr9_P_ P atom and the U1 and U2 atoms were 6.99 Å and 11.67 Å for site I and site II, respectively, and the N-Cα-Cβ-Oγ dihedral angle of Thr9_P_ was −164.7°. Looking for a new conformation where the phosphoryl group would interact with uranyls, NOE-type distance restraints were applied using a bi-harmonic potential with a force constant of 20 kcal mol^−1^ Å^−2^ between the site I Thr9_P_ P atom and uranyl U1 atom of site I (above 3.0 and below 5.0 Å) as well as uranyl U2 atom of site II (above 3.0 and below 9.0 Å)

The system was energy minimized and subject to 200 ps Langevin dynamics at 300 K to yield the structure “With-nOe”. After the restrained distance dynamics, distances between P and U decreased to 5.18 Å and 9.40 Å, respectively, for site I and site II and the N-Cα-Cβ-Oγ dihedral angle of site I Thr9_P_ was −92.5°.

The Amber program was then used to run production dynamics runs using a polarizable force field [46]. The previous *NonOe* and *With-nOE* systems were translated into Amber format with the ff14SB force field [47]. A TPO residue was used for site I Thr9_P_. The uranyl parameters used for the polarizable MD simulations are taken from [48]. The Amber systems were energy minimized first without polarization and then again with polarization, which was used for the rest of the equilibration and production. The equilibration consisted of 100 ps NPT (constant pressure and temperature *τ_P_* = 2.0 ps) dynamics from the minimum energy structure followed by 100 ps NPT dynamics, all with polarization. A time step of 1 fs, a temperature of 300 K using Berendsen control and a cutoff for the non-bonded interactions set to 12 Å were employed. Particle Mesh Ewald was used for electrostatics with Car–Parinello scheme for dipoles, dipole temperature constant *dipτ* = 9.9 ps and order 5 for B-spline interpolation. Finally, 12 ns NPT MD simulations were run with the same conditions.

## 3. Results

### 3.1. Design and Characterization of the Calmodulin Variants

To study the binding properties of uranyl to CaM site II, the effect of cooperativity between binding sites I and II of the CaM N-ter domain and the joint effect of cooperativity and phosphorylation at site I, eleven mutants were produced (Table 1). Five “one-site” peptides with site II only able to bind metal ions were produced. The numbering used in this study corresponds to the amino acids in each metal-binding loop, ranging from 1 to 12. In these mutants, metal binding at site I was impaired by introducing the Asp1Ala and Asp3Ala mutations at the site I binding loop. In the six “two-site” peptides, both sites I and II were able to chelate a metal ion. The CaM1 Y I-II and CaM1 YW I-II peptides bear the TAAE sequence in site I, necessary for site I Thr9 phosphorylation. They were studied as reference peptides for the CaM1 Y I-II P and CaM1 YW I-II P peptides, which correspond to the “two-site” peptides with Thr9_P_ at site I.

Site I Tyr7 is an efficient probe of uranyl binding at both sites I and II, since we observed a decrease in its fluorescence emission (302 nm) upon uranyl binding at site II in the CaM Y II and CaM1 Y II peptides and at site I and site II in the two-site peptides. For the peptides having a tryptophan at position 7 of site II, CaM YW II, CaM1 YW II, CaM YW I-II, CaM1 YW I-II and CaM1P YW I-II (Table 1), the metal-binding affinity of site II and site I was assessed following the dominating tryptophan fluorescence emission at 350 nm. The effect of the site II Thr7 to Trp mutation on the microscopic binding constants and cooperativity was also analyzed.

### 3.2. Uranyl-Binding Properties of Site II Variants of Calmodulin N-Terminal Domain

Spectrofluorimetric titrations of the site II variants (one-site peptides) with UO_2_^2+^ were satisfactorily fit using binding isotherms, which represent the formation of 1:1 complexes (Figure 2 and Appendix A). All the investigated peptides show a fluorescence emission decrease upon addition of UO_2_^2+^. The spectra collected during titrations are reported in Appendix A. Titrations were performed in the presence of iminodiacetate (IDA), used to control uranyl speciation (see Section 2, [16]). The macroscopic binding constants obtained for all the investigated peptides are reported in Table 2, together with the macroscopic constants for Ca^2+^ binding to the same peptides, evaluated in a previous work [28].

For the one-site peptides, since only site II is able to bind the metal ion, the measured macroscopic constants correspond to the microscopic constants of the active binding site (K_1_ = K_II_).

The binding affinity of site II for uranyl is one order of magnitude lower than that of site I with K_II_ = (3.7 ± 0.5) × 10^6^ M^−1^ and K_I_ = (4.0 ± 0.2) × 10^7^ M^−1^, respectively (corresponding to Kd_II_ = 270 ± 42 nM and Kd_I_ = 25 ± 1 nM, respectively). The uranyl-binding affinity of site II is not significantly affected by the site I Thr10Ala and Lys11Ala mutations, with a value of K_II_ = (3.5 ± 0.7) × 10^6^ M^−1^ for CaM1 Y II (Kd_II_ = 286 ± 47 nM).

The affinity of site II for uranyl slightly increases in the peptides containing the mutation Thr7Trp in site II, with the microscopic stability constant K_II_ = (5.1 ± 0.5) × 10^6^ M^−1^ for CaM YW II and K_II_ = (5.9 ± 0.6) × 10^6^ M^−1^ for CaM1 YW II, respectively (Kd_II_ = 196 ± 21 and 169 ± 20 nM, respectively; see Table 2). 

Isothermal titration calorimetry (ITC) measurements of UO_2_^2+^ binding to the one-site peptides provided the enthalpy and entropy values reported in Table 3. Uranyl-binding to site II was found to be exothermic (Figure 3A,B). The binding isotherms saturated near a 1:1 stoichiometric ratio and data were well described by a one-site binding model (see the paragraph ‘Calculations’ in the Section 2).

The values of the macroscopic constant K_1_ were in good agreement with those obtained from spectrofluorimetric analysis (Table 2). The enthalpy change associated with uranyl-binding was significantly higher for peptides with the site II Thr7Trp mutation (Table 3). On the contrary, the difference in ΔH values due to site I Thr10Ala and Lys11Ala mutations was negligible or very small: similar enthalpy values were obtained for CaM Y II and CaM1 Y II (−2.0 ± 0.9 and −3.5 ± 1.0 kJ mol^−1^), and for CaM YW II and CaM1 YW II (−8.9 ± 0.7 and −15.0 ± 2.5 kJ mol^−1^). The entropic term for each uranyl–peptide interaction was calculated using Equation (14). As previously pointed out for Ca(II) binding to one-site peptides [28], in the case of uranyl binding to site II, the absolute values of ΔH are also small and the enthalpic contribution to the Gibbs free energy is about 10 times lower than the entropic term, with TΔS values ranging from 24.7 to 35.8 kJ mol^−1^. 

### 3.3. Two-Site Peptides and the Cooperative Effect

Spectrofluorimetric isotherms for UO_2_^2+^ binding to two-site peptides were analyzed using a dinuclear model, providing two macroscopic constants, K_1_ and K_2_ (Figure 1C, Table 2). Figure 2C,D shows the experimental data and the fitting curves obtained from Equation (4). These macroscopic constants are from two to three orders of magnitude higher for uranyl than for calcium. In addition, as observed for the one-site peptides, both K_1_ and K_2_ assume slightly higher values when the point mutation Thr7Trp is inserted in site II. 

To quantitatively assess a possible inter-site cooperativity related to UO_2_^2+^ binding, we evaluated the free energy of site–site interaction, ΔΔG (Table 4). For this purpose, the evaluation of microscopic constants is necessary (cf. the paragraph ‘Calculations’ in the Section 2).

In first approximation, we introduced the K values obtained for the one-site peptides binding UO_2_^2+^ at site II (CaM Y II and CaM YW II, CaM1 Y II and CaM1 YW II) as the microscopic constants K_II_ in Equations (5) and (6), together with the experimental macroscopic K_1_ and K_2_ constants obtained for the corresponding two-site peptides. This strategy is justified by the fact that the substituted residues at site I Ala1 and Ala3, which impair metal binding at site I, were shown not to be involved in the site I–site II interaction [49]. In support of this assumption, circular dichroism experiments previously showed equivalent conformations for the CaM N-ter domain with and without the site I mutations Asp1Ala and Asp3Ala [28]. Consequently, the mononuclear binding process at site II was considered equivalent for the one-site and the corresponding two-site variants. Importantly, the microscopic constants K_I_ thus calculated for CaM Y I-II and CaM1 Y I-II (3.1 ± 0.3 × 10^7^ M^−1^ and 2.7 ± 0.3 × 10^7^ M^−1^) are consistent with the K_I_ previously measured for the corresponding one-site peptides CaM Y I and CaM1 Y I (K_I_ = 4 ± 0.2 × 10^7^ M^−1^ and 3.1 ± 0.1 × 10^7^ M^−1^, respectively [16]). 

A positive cooperative effect (ΔΔG < 0) is associated with UO_2_^2+^ binding to the CaM N-ter domain, with ΔΔG = −2.0 and −1.8 kJ mol^−1^ for UO_2_^2+^ binding to CaM Y I-II and CaM YW I-II, respectively. Consequently, the affinity of the binding sites increases when one metal ion is already bound to the peptide (K_I,II_ > K_I_ and K_II,I_ > K_II_). Even if this positive cooperative effect is evident, the ΔΔG obtained for UO_2_^2+^ binding is three times lower than that obtained for Ca^2+^ binding to the same peptides (ΔΔG = −6.1 and −6.9 kJ mol^−1^, respectively [28]).

ITC measurements provided negative total heat values for the four variants of CaM (Figure 3C,D and Appendix A). The binding isotherms were adjusted to a two-site cooperative binding model, which takes into account the microscopic binding constants and enthalpies for the two sites, together with an enthalpy associated with the cooperative effect, ΔH_c_ (see the paragraph ‘Calculations’ in the Section 2). Equation (13) was used to adjust the experimental data. As illustrated above, the binding process at site II of the two-site peptides could be considered equivalent to the binding process at site II of the corresponding one-site mutants. Therefore, we replaced K_II_ and ΔH_II_ of the two-site peptides with the K_II_ and ΔH_II_ values obtained for the corresponding one-site peptides. In this way, Equations (5) and (13) allowed us to evaluate K_I_, ΔH_I_ for site I, the macroscopic constants K_1_ and K_2_ and, finally, the enthalpic contribution of the inter-site cooperativity, ΔH_c_ (Table 3 and Table 4).

The obtained macroscopic constants are in good agreement with spectrofluorometric results (Table 2). Results reported in Table 3 reveal that binding at site I is exothermic and more favorable than binding at site II with a larger enthalpic contribution, for all the investigated peptides. Absolute values of ΔH_I_ become slightly lower in CaM1, i.e., when the Thr10Ala and Lys11Ala point mutations are inserted at site I, but this has no influence on the enthalpy associated with the cooperativity. In fact, ΔH_c_ is similar for CaM Y I-II and CaM1 Y I-II (25.0 ± 0.1 and 22.0 ± 0.2 kJ mol^−1^). For CaM YW I-II and CaM1 YW I-II, the difference between ΔH_I_ and ΔH_II_ is smaller, because ΔH_II_ is slightly higher, and the enthalpy associated with cooperativity is much smaller, as compared to peptides without the tryptophan with ΔH_c_ = 3.8 ± 0.1 and 8.9 ± 0.3 kJ mol^−1^ (Table 3). According to these data, the enthalpy associated with cooperativity is unfavorable (positive values), especially for the peptides that do not have the tryptophan residue at site II.

Entropy values were evaluated from the corresponding ΔH and equilibrium constants, using Equation (14). For uranyl binding at site I, entropic terms (TΔS_I_) ranging between 3.2 kJ mol^−1^ K^−1^ and 44.5 kJ mol^−1^ K^−1^ were obtained. The calculated values of ΔH_I_ and the enthalpic contribution to the Gibbs free energy are larger than the entropic term, except for the two-site peptide CaM1 YW I-II (Table 3). The highest values of TΔS_I_ were obtained for the peptides presenting the tryptophan residue at site II, showing that the influence of this residue on uranyl binding at site I, already pointed out as having the highest equilibrium constants, is linked to entropic factors. Such an effect is possibly due to interactions between Trp7 at site II and Tyr7 at site I. Indeed, it was previously shown that, within these peptides, Trp7 (site II) and Tyr7 (site I) are part of an inter site β-sheet structure, which can tune the interaction between the main chain carbonyl of both the residues and the metal ion [28] (Figure 1B). In contrast, the values of TΔS_II_ were almost equivalent in all the variants, and only slightly lower in the peptides bearing the tryptophan residue (Table 3). They were significantly higher than the TΔS_I_ values, except for the CaM1 YW I-II peptide. The entropic term for the inter-site cooperativity (TΔS_c_) is favorable for all the investigated peptides. This result is consistent with the assumption that CaM mutants undergo conformational changes upon binding the first uranyl ion, which favor the binding of a second ligand, which happens as well for Ca^2+^ binding to CaM [28]. 

### 3.4. Influence of Site I Thr9 Phosphorylation on Uranyl Binding

Spectrofluorimetric measurements reported in Figure 4 show that phosphorylation of site I Thr9 (Thr9_P_) increases the affinity for uranyl of both sites I and II. At pH 6, we obtained macroscopic stability constants for the phosphorylated variants from 5- to 14-fold higher than the constants of the corresponding non-phosphorylated peptides (Table 2). Site I phosphorylation increases the uranyl affinity of site II also when site I is inactivated. In CaM1 Y II P, a one-site peptide with site I Thr9_P_ and inactivated for metal binding at site I, spectrofluorimetric titrations with uranyl provide a macroscopic constant K_1_ = K_II_ = (2.1 ± 0.4) × 10^7^ M^−1^ at pH 6 (Kd = 48 ± 11 nM, titration curve in Appendix A). This value is *circa* fivefold higher than that of the corresponding non-phosphorylated peptide CaM1 Y II, with K_II_ = (3.7 ± 0.5) × 10^6^ M^−1^ (Kd = 270 ± 42 nM). At pH 7, the affinity of CaM1 Y II P for uranyl further increases, with K_II_ = (1.8 ± 0.4) × 10^8^ M^−1^ (Kd = 5.5 ± 1 nM, Table 2). Since direct interaction of the phosphoryl group with uranyl at site II is not possible, this behavior points a structural reorganization mediated by site I phosphorylation that strongly influences site II affinity for uranyl.

The microscopic constants and ΔΔG for the two-site phosphorylated peptides were evaluated using their macroscopic constants, K_1_ and K_2_, and the K_II_ of the one-site peptide, CaM1 Y II P, following the method used for the non-phosphorylated peptides. It is remarkable that for the CaM1 Y I-II P peptide, the calculated K_I_ value of (3.4 ± 0.3) × 10^8^ at pH 6 is very close to the experimental value of (2.0 ± 0.4) × 10^8^ previously reported for the CaM1 Y I P peptide [16].

CaM1 Y II P does not have the site II Thr7Trp mutation, but its K_II_ value was used also for the evaluation of the microscopic constants of CaM1 YW I-II P. As detailed above, the effect of site II Trp7 insertion on uranyl binding (K_II_ 1.4 times higher with site II Trp7) is low compared to the effect of site I phosphorylation (K_II_ circa five times higher). Therefore, we consider that the CaM1 Y II P macroscopic constant could represent a good estimation of K_II_ for CaM1 YW I-II P as well.

The ΔΔG values obtained for the two phosphorylated peptides show a significant enhancement of cooperativity in the presence of the phosphoryl group. The ΔΔG values for CaM1 Y I-II P and CaM1 YW I-II P are (−3.4 ± 0.4) and (−4.7 ± 0.4) kJ mol^−1^, respectively, as compared to (−2 ± 0.4) and (−1.8 ± 0.3) kJ mol^−1^ for the non-phosphorylated peptides.

The binding affinity of uranyl for CaM1 Y I-II P and CaM1 YW I-II P was further improved at pH 7 (Figure 4C,D) with the macroscopic constants K_1_ two orders of magnitude higher, and K_2_ almost 10-fold higher, as compared to pH 6 (Table 2). For CaM1 Y I-II P, these macroscopic constants correspond to microscopic dissociation constants for the MP and M_2_P complexes that lie in the picomolar range for uranyl binding at site I, with Kd_I_ = 10 pM and Kd_I,II_ = 2.56 pM (scheme in Figure 1C). The microscopic dissociation constants are in the nanomolar range for uranyl binding at site II, with Kd_II_ = 5.5 nM and Kd_II,I_ = 1.4 nM. At pH 7, the dissociation constants at site I thus significantly differ in the two-site peptide and in the one-site peptide CaM1 Y I P, for which Kd_I_ = 320 ± 54 pM [16]. We cannot completely exclude the possibility that K_II_ could also be larger in the CaM1 Y I-II P peptide as compared to CaM1 Y II P. However, the very high value of K_1_ cannot be reconciled by a value of K_I_ close to that of the one-site CaM1 Y I P peptide, even by increasing the value of K_II_ tenfold. In addition, an increase in pH from pH 6 to pH 7 has no effect on the cooperativity of uranyl binding to CaM1 Y I-II P (Table 4). The experimental data point an effect of pH and phosphorylation, which affects the structure of site I differently in the two-site CaM1 Y I-II P peptide and in the one-site CaM1 Y I P peptide.

For CaM1 YW I-II P, similar conclusions can be proposed, although the value of ΔΔG approximately doubles at pH 7. This value depends, however, on the microscopic K_II_ constant used for the calculation and may therefore not be reliable. Notably, the two uranyl macroscopic binding constants of CaM1 YW I-II P, K_1_ = (6.0 ± 0.9) × 10^11^ and K_2_ = (5.1 ± 0.6) × 10^9^ correspond to dissociation constants in the picomolar to sub-nanomolar range, with Kd_1_ = 1.7 ± 0.3 pM and Kd_2_ = 196 ± 21 pM, respectively. The two phosphorylated two-site variants, CaM1 Y I-II P and CaM1 YW I-II P, thus present a very high affinity for uranyl at pH 7.

### 3.5. Ca^2+^ Binding to the Peptide Variants

Since calmodulin is a calcium-binding protein, calcium may reasonably be the main competitor of uranyl in binding to CaM sites in vivo. We checked, for comparison, the affinity of the phosphorylated peptides for Ca(II) by performing spectrofluorimetric titrations of CaM1 Y I-II P and CaM1 YW I-II P with a CaCl_2_ solution (binding isotherms are presented in Appendix A). We observed a negligible effect of phosphorylation on the affinity for calcium at pH 6, and a very modest one at pH 7, for CaM1 Y I-II P (Table 2). For CaM1 YW I-II P, there are also only negligible effects of phosphorylation and of pH on the affinity for Ca(II). All studied peptides show affinities in the 5 × 10^4^ to 2.8 × 10^5^ M^−1^ range, corresponding to dissociation constants between 3.6 µM and 20 µM, indicating a high uranyl vs. calcium selectivity of the two-site peptides.

### 3.6. Structural Modelization of CaM1 Y I-II P Using Molecular Dynamics

To better understand the influence of phosphorylation on the affinity of uranyl to both site I and site II and the strong increase in affinity at site I of the phosphorylated two-site peptide at pH 7, as compared to phosphorylated site I only, we have modelled the complex formed by CaM1 Y I-II P with two UO_2_^2+^ in water. As detailed in the Section 2, two structural models were considered that differ in the distance of the phosphoryl group of site I Thr9_P_ to the uranium atoms U1 and U2 of site I and site II, respectively. Large distances were obtained in a structure without constraints. Shorter P-U distances and a different N-Cα-Cβ-Oγ dihedral angle for site I Thr9_P_ were obtained in a structure with NOE-type distance restraints applied between the Thr9_P_ P atom and U1 and U2. Production dynamics were run with the Amber program, starting with the structures without constraints (“No-Restr”) or with constraints on the initial distance P-U1 and P-U2 (“With-Restr”).

The polarization energy was the term, which was the longest to equilibrate, with a stabilization after *circa* 6 ns and the total energy followed the same evolution (Figure 5A,B). The coordinate RMSD of backbone atoms was very small (1.11 Å and 0.87 Å on average after 6 ns for the “No-Restr”. and “With-Restr.” Systems, respectively; see Figure 5C) and the structures were very well conserved as seen by the final MD frames represented in Appendix A. Graphs showing the evolution of the P-U1 and P-U2 distances and the N-Cα-Cβ-Oγ dihedral angle of site I Thr9_P_ are shown in Appendix A and the geometric parameters given in Appendix A.

These results show that the CaM1 Y I-II P-(UO_2_^2+^)_2_ system is stable in two different conformations, with different rotamers of site I Thr9_P_. The structures after 12 ns MD of the “No-Restr.” and “With-Restr.” systems are compared in Figure 6. Residues with at least one atom closer than 2.5 Å to the uranyl molecules are the same in the two structures but the phosphoryl group of site I Thr9_P_ is clearly closer to the U atoms in the “With-Restr.” System, corresponding to the second rotamer mentioned above. This structure could explain, at least in part, the experimental results showing an effect of the phosphoryl group on both site I and, to a lesser extent, site II. Interestingly, the inter-site β-strand motif is longer in the “With-Restr.” simulation (Figure 6B) and we witness a slightly more stable structure in the nanosecond time range. This may be explained by the presence of a water molecule bridging one phosphoryl oxygen of site I Thr9_P_ and the peptide carbonyl of site II Gly6 as shown in Appendix A. Notably, the coordination sphere of U1 at site I does not involve the phosphoryl group. It is composed of five oxygen atoms well distributed in the equatorial plane of uranyl, with monodentate site I Asp3 and Asp5, the Tyr7 carbonyl oxygen and bidentate Glu12. For site II, the U2 coordination sphere is also composed of five oxygen atoms in the equatorial plane, with site II monodentate Asp3 and Asp9, bidentate Glu12 and a water molecule Table 5.

## 4. Discussion

The aim of this work was to study the binding affinity of uranyl for the CaM site II and the extent of uranyl-binding cooperativity at sites I and II in the CaM N-ter domain. We also aimed at using the CaM N-ter domain as template to produce variants on the two metal-binding sites and modulate their affinity for uranyl.

### 4.1. Site II Affinity for Uranyl

We showed that the CaM site II has a 10-times lower affinity for uranyl than site I, with K_II_ = 3.7 × 10^6^ M^−1^, compared to K_I_ = 4 × 10^7^ M^−1^ for the one site peptide CaM Y I [16]. This corresponds, however, to a dissociation constant in the nanomolar range with Kd_(site II)_ = 270 ± 42 nM, which is 200 times lower than the dissociation constant measured for calcium (Kd_(site II)_ = 55 µM [28]). For site I, a structural model obtained by MD simulations proposed a coordination of uranyl with five equatorial ligands, i.e., bidentate Asp3, monodentate Asp5 and Glu12, and the Tyr7 carbonyl, with distances O-U1 between 2.42 Å and 2.52 Å [34]. These MD simulations were performed in *vacuum* and indeed, time-resolved laser-fluorescence spectroscopy and extended X-ray-absorption fluorescence spectroscopy strongly indicated the presence of a hydroxyl ligand [34]. The sequence of the site II binding loop differs significantly from that of site I, with an Asn at position five and an Asp at position nine of the loop. The lower affinity of site II for uranyl may be due to these differences in coordination sphere geometry. In the structural model obtained by MD simulations on the two-site CaM1 Y I-II P peptide, the coordination sphere of U2 at site II involves a water molecule and Asp9 as a monodentate ligand, in addition to monodentate Asp3 and bidentate Glu12 (Table 5). There is no involvement of the residues at position five or seven of the metal-binding loop, and hence no direct interaction of residues involved in the inter-site β-sheet structure.

### 4.2. “Two-Site” N-Ter Domain

A notable improvement of the CaM N-ter domain ability to bind uranyl is observed when both binding sites are activated. Obviously, in this case, each peptide is able to bind two uranyl ions instead of one, but on top of that, a cooperative effect increases the affinity of each binding site, with K_I,II_ > K_I_ and K_II,I_ > K_II_ for all the investigated peptides. To the best of our knowledge, only few studies have been performed on metals’ cooperative binding to calmodulin or calmodulin-derived peptides [27,32,50,51,52]. The cooperative effect related to their binding was only qualitatively measured, by means of the Hill equation. In this work, the inter-site cooperativity was quantitatively evaluated for all the investigated two-site peptides, which allowed us to correlate this effect with the peptide molecular structure and the point mutations introduced into the metal-binding loops.

A positive cooperative effect is observed for uranyl binding to the non-phosphorylated peptides, even if it is lower than that observed for calcium binding. The ΔΔG parameter for CaM Y I-II and CaM YW I-II lies between −2.0 and −1.3 kJ mol^−1^, while we previously found ΔΔG = −6.1 kJ mol^−1^ and ΔΔG = −6.9 kJ mol^−1^, respectively, for calcium binding to these peptides [28]. The cooperative effect strongly depends on the nature of the metal ion. Studies on Ca^2+^ and Mg^2+^ binding to a peptide containing sites II and III of CaM showed that the α-helical content of the peptide increased by 12–24% in the presence of Ca^2+^, while no conformational change was observed for Mg^2+^ binding [53]. The stabilization of α-helical conformation, upon uranyl binding, was previously observed by Le Clainche and Vita for a CaM modified site I, which differs from site I of our peptides for the Asp1Thr and Asp5Thr mutations [54].

The conditional constants recently reported for calcium binding [28] are at least two orders of magnitude lower than those obtained in the present work for the interaction of the same peptides with uranyl. This result represents a first step towards engineering peptides with high affinity and selectivity for uranyl.

A modest contribution to the improvement of uranyl affinity for CaM N-ter domain is given by the insertion of a tryptophan residue at position seven of site II. In fact, our results show that this mutation slightly increases the affinity of both sites of the CaM N-ter domain for uranyl. Both microscopic constants, K_I_ and K_II_, are almost twofold higher for the peptides having the site II Thr7Trp mutation, compared to the corresponding peptides without this mutation. This small affinity improvement for both sites was already observed for calcium binding to the same peptides [28] and it was attributed to the higher hydrophobicity of tryptophan when compared to threonine, which could facilitate the folding of the binding loop and reduce the distances between the binding amino acids and the metal ion. We have an affinity improvement of the same order of magnitude for uranyl as for calcium, which suggests that this effect is probably independent on the metal ion. Both affinity and selectivity are further improved when coupling the cooperative effect and phosphorylation of site I Thr9, as detailed below. 

Calorimetric measures provided further information on the thermodynamics at the microscopic level of uranyl binding to the calmodulin N-ter domain. Both bindings at site I and site II are exothermic (ΔH_I_ and ΔH_II_ < 0). However, only binding at site II is clearly entropy driven, showing small values of ΔH_II_ compared to TΔS_II_. The ΔH_II_ values for uranyl binding are similar to those obtained for calcium binding to the same site [28]. The positive entropy change at site II can be due to both hydrophobic interactions (especially for mutants containing site II Trp7) or metal–protein electrostatic interactions.

Absolute values of enthalpy for binding at site I are higher than those obtained for site II. More negative enthalpy values at site I can be explained with the higher number of charged amino acids in this site, which favors hydrogen bonding and Van der Waals interactions, both associated with negative enthalpy changes [55]. 

For mutants with the site II Thr7Trp mutation, contributions of ΔH_I_ and TΔS_I_ to the total Gibbs free energy are comparable, probably because site II Trp7, with its hydrophobic side chain, may interact with the Tyr residue at site I, increasing the positive entropy change associated with the hydrophobic contribution to the binding reaction [55]. 

### 4.3. Phosphorylated Site I Considerably Improves Uranyl Binding at Site I and Site II in the N-Ter Domain

The most significant improvement of both mononuclear (MP) and dinuclear (M_2_P) complex stability is obtained with site I Thr9 phosphorylation. We previously showed, by FTIR and EXAFS analysis of the one-site peptide CaM1 Y I P, that the phosphoryl group is directly involved in uranyl coordination [16,34] and that it is completely deprotonated in the dianionic form -OPO_3_^2-^ at pH 7. 

In the case of the two-site peptide CaM1 Y I-II P, the uranyl–peptide complex stabilization due to site I phosphorylation is combined with a positive inter-site cooperativity, leading to a synergistic effect that results in picomolar microscopic dissociation constants for uranyl binding at site I and nanomolar ones for dissociation constants at site II. However, our results show that the influence of phosphorylation on uranyl binding is not linked to its direct coordination to the metal ion. Concerning site I, the affinity deduced from the microscopic constant K_I_ at pH 6 is similar to that measured for the one-site peptide CaM1 Y I P. In contrast, at pH 7, K_I_ shows a strong increase in affinity for uranyl. It is larger than that recorded with the one-site CaM1 Y I P peptide at the same pH by more than two orders of magnitude with 1.0 ± 0.5 × 10^11^ vs. 3.12 × 10^9^, corresponding to dissociation constants Kd = 10 ± 8 pM vs. Kd = 320 ± 47 pM. These results imply that for a phosphorylated site I, a different coordination sphere exists for the one-site and for the two-site peptides CaM1 Y I P and CaM1 Y I-II P. Since the peptides differ by site II Asp1Ala and Asp3Ala mutations, these differences must be associated with specific interactions involving the deprotonated phosphoryl group and residues at the beginning of the site II binding loop. Such an interaction is also supposed to influence site II binding properties. This is discussed below. Concerning the high uranyl affinity for site I observed without the presence of the strong phosphoryl ligand, it may be due in part to an optimized structural organization of the binding site, with the five well-distributed equatorial ligands, and to the participation of site I Glu12 as a bidentate ligand. Indeed, bidentate coordination by Glu12 has been shown to play an important role in closing the calcium binding in the EF-hand calcium-binding motifs and to reorient the flanking alpha helices [56]. Bidentate uranyl coordination by this Glu was also proposed to participate in the high stability of the uranyl complex formed by a CaM site I variant with a 10-amino-acid long uranyl-binding loop [38].

The K_II_ values obtained for the phosphorylated one-site peptide CaM1 Y II P also show that phosphorylation at site I can have an important effect on uranyl binding without direct coordination. At pH 6, the affinity of site II in CaM1 Y II P is six times higher than the non-phosphorylated peptide CaM1 Y II. At pH 7, the affinity for site II in CaM1 Y I-II P further increases by a factor of 8.6, with a dissociation constant Kd = 5.5 nM. 

The insertion of a phosphoryl group can lead to protein conformational changes that improve the affinity at the two sites. Such an effect may also explain the higher inter-site cooperativity ΔΔG for phosphorylated peptides at pH 6 as compared to the non-phosphorylated peptides (ΔΔG = −3.4 ± 0.4 kJ mol^−1^ vs. −2 ± 1 kJ mol^−1^, Table 4). In particular, a larger β-sheet structure linking site I and site II is proposed in the structural model of CaM1 Y I-II P obtained by MD simulations (Figure 6B). The presence of a hydrogen-bonding network involving a water molecule between the phosphoryl group of site I Thr9_P_ and site II Gly6 carbonyl (Figure SI-11) may extend the inter-site antiparallel β-sheet motif that involves residues at positions seven and eight of the metal-binding loops. The change in uranyl-binding cooperativity may be explained by this structural difference, which could modify the peptide secondary structure upon first metal binding, leading to a more accessible conformation for binding a second metal ion.

The phosphorylated peptides, besides their high affinity for uranyl ions, have also proved to be highly selective for this metal ion over Ca^2+^, with conditional macroscopic binding constants that are three orders of magnitude higher for UO_2_^2+^ than for Ca^2+^ at pH 6 and up to six orders of magnitude higher at pH 7.

## 5. Conclusions

In this work, we have studied the interaction properties of uranyl with engineered N-ter domains of the regulatory calcium-binding protein calmodulin (CaM). By site-directed mutagenesis, we could produce CaM variants with one and two uranyl-binding sites and evaluate both macroscopic and microscopic (site-specific) thermodynamic parameters associated with uranyl binding to site I and site II of the N-ter domain. This strategy allowed us to modulate the affinity of the CaM N-ter domain for uranyl, by identifying structural factors governing this interaction at the molecular level. We showed that the phosphorylation of Thr9 at site I increases the natural inter-site cooperativity of calmodulin N-ter domain, and leads to peptides able to strongly complex uranyl, with picomolar dissociation constants and a high selectivity towards the cognate ion calcium. 

These findings give further insights into the principles of uranium interaction with proteins, and its possible toxicological effect on biological targets. Our approach could also be a starting point for developing uranyl-selective biological tools, based on mutated peptides, for bioremediation and biodetection purposes.

## Figures and Tables

**Figure 1 biomolecules-12-01703-f001:**
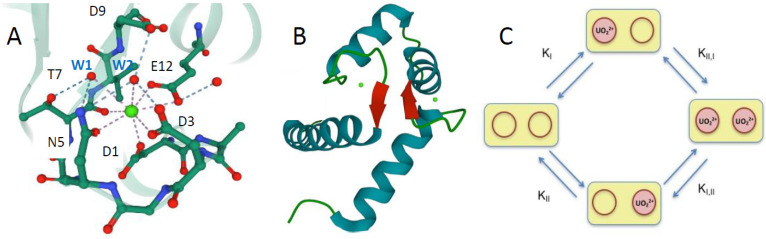
(**A**) Scheme of the calcium-binding site II of CaM of *Paramecium tetraurelia* (PDB 1EXR). Coordination of the Ca^2+^ ion (in green) involves monodentate Asp1, Asp3 and Asn5 (D1, D3, and N5), the carbonyl of Thr7 (T7), bidentate Glu12 (E12) and two water molecules (W1 and W2) stabilized by hydrogen-bonding interactions (dashed blue lines); (**B**) Scheme of the N-ter domain of CaM with the intersite betasheet structure highlighted in red and calcium ions in green. (**C**) Schematic diagram of the UO_2_^2+^ binding pathways in the CaM N-ter domain, red circles represent the metal-binding sites. Microscopic binding constants are indicated for each binding step.

**Figure 2 biomolecules-12-01703-f002:**
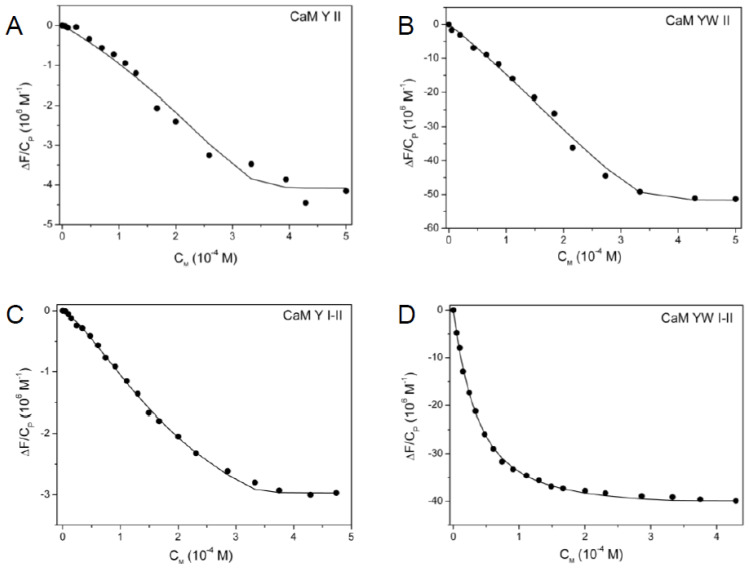
Fluorescence binding isotherms for the interaction of UO_2_^2+^ with the one-site peptides (**A**) CaM Y II (at λ = 302 nm) and (**B**) CaM YW II (at λ = 350 nm), the two-site peptides (**C**) CaM Y I-II (at λ = 302 nm) and (**D**) CaM YW I-II (at λ = 350 nm); initial C_P_ = 1.10^−5^ M, initial C_I_ = 5.10^−4^ M, pH = 6, T = 298 K. Experimental data (dots) and adjustment (black solid line), according to Equation (5) for (**A**,**B**) and Equation (7) for (**C**,**D**). Analogue binding isotherms are reported in Appendix A for CaM1 Y II, CaM1 Y II P and CaM1 YW II and in Appendix A for CaM1 Y I-II and CaM1 YW I-II.

**Figure 3 biomolecules-12-01703-f003:**
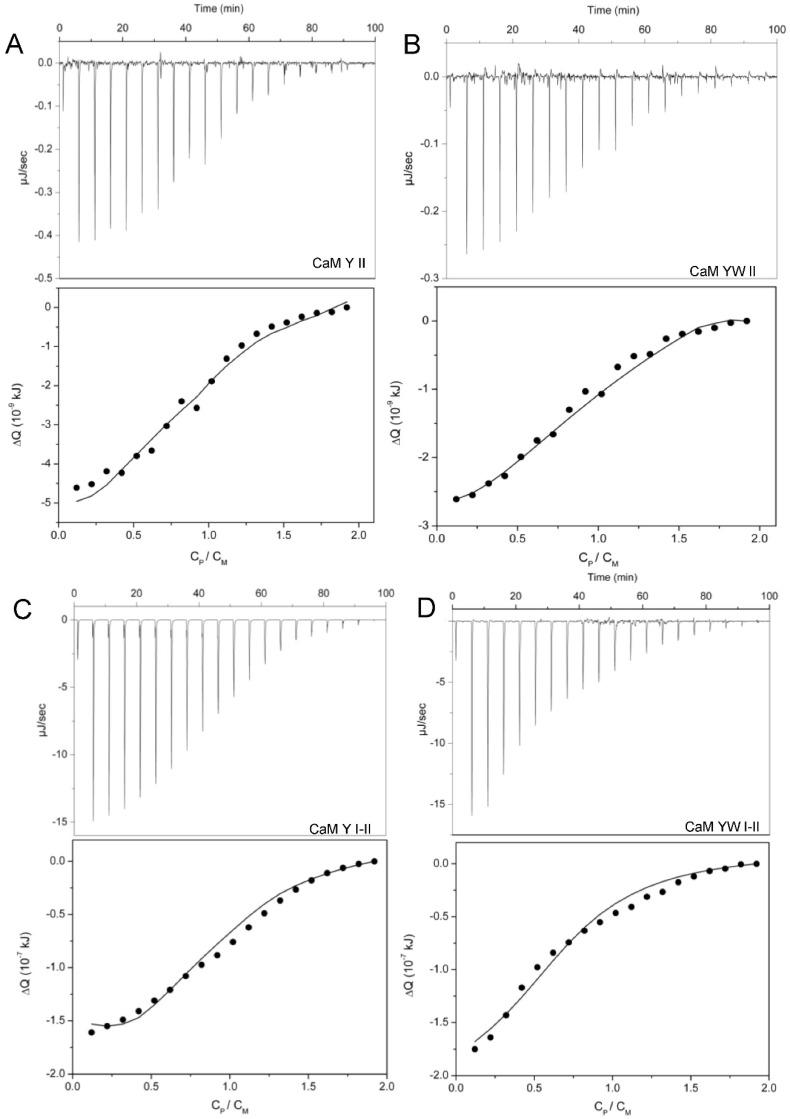
ITC raw data and binding isotherm for UO_2_^2+^ interaction with the one-site peptides (**A**) CaM Y II and (**B**) CaM YW II; C_P_ = 1.10^−3^ M (syringe), C_I_ = 2.10^−4^ M and C_M_ = 1.10^−4^ M (cell) and with the non-phosphorylated two-site peptides (**C**) CaM Y I-II and (**D**) CaM YW I-II; C_P_ = 2.10^−3^ M (syringe), C_I_ = 4.10^−4^ M and C_M_ =2.10^−4^ M (cell). pH = 6, T = 298 K. Experimental data (dots) and adjustment (black solid line), according to Equation (8) for the one-site peptides and Equation (13) for the two-site peptides. Analogue binding isotherms are reported in Appendix A for CaM1 Y II and CaM1 YW II and in SI-6 for CaM1 Y I-II and CaM1 YW I-II.

**Figure 4 biomolecules-12-01703-f004:**
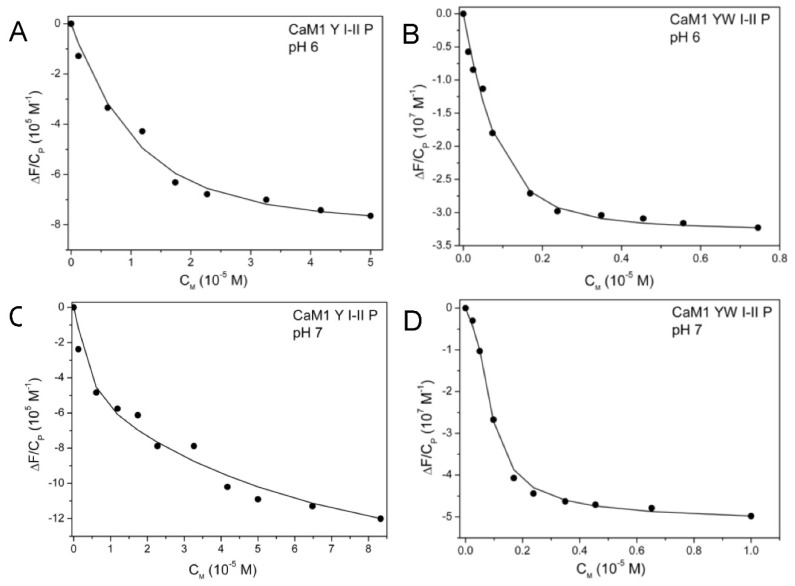
Fluorescence binding isotherms for the interaction of UO_2_^2+^ ion with the phosphorylated two-site peptides CaM1 Y I-II P (at λ = 302 nm) at pH 6 (**A**) and pH 7 (**C**) and CaM1 YW I-II P (at λ = 350 nm) at pH 6 (**B**) and at pH 7 (**D**). Initial C_P_ = 5.10^−6^ M, initial C_I_ = 5.10^−4^ M for CaM1 Y I-II P, initial C_P_ = 5.10^−7^ M, initial C_I_ = 5.10^−5^ M for CaM1 YW I-II P. T = 298 K. Experimental data (dots) and adjustment (black solid line), according to Equation (4).

**Figure 5 biomolecules-12-01703-f005:**
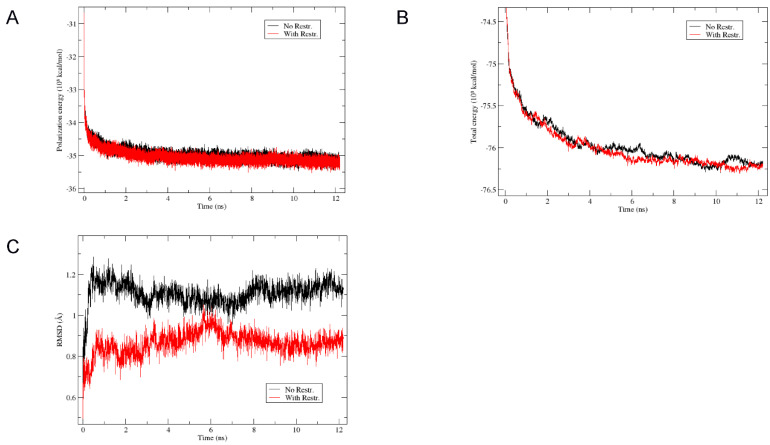
Evolution of (**A**) the polarization energy, (**B**) the total energy, and (**C**) the RMSD of backbone atoms during the 12 ns NPT MD simulation performed with the Amber program using the ff14SB force field without (“No Restr.”, black curve) or with (“With Restr.”, red curve) constraints on the initial P-U1 and P-U2 distances.

**Figure 6 biomolecules-12-01703-f006:**
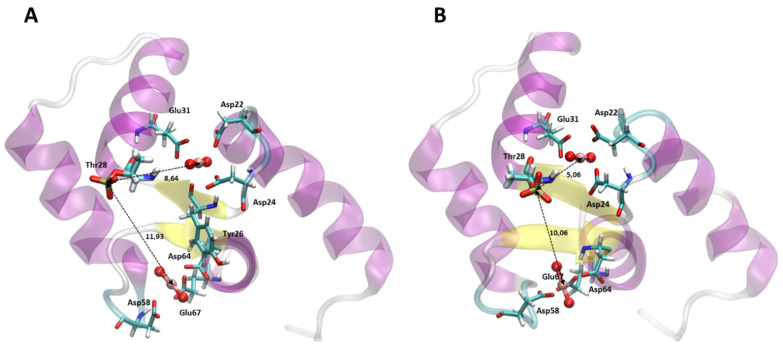
Structures of the calmodulin CaM1 Y I-II P system obtained after 12 ns MD simulation with a polarized force field (**A**) without restraints (**B**) following the introduction of restraints before starting the simulation. Residues with at least one atom closer than 2.5 Å to the uranyls (in CPK) are represented in licorice. Distances between the P atom of the phosphorylated threonine and uranium atoms are also given. The numbering used in this figure is that of the global N-ter domain, composed of 77 amino acids. Here is the correspondence referring to the numbering of metal-binding sites used in this article: Asp22 corresponds to Asp3 of site I, Asp24 to Asp5 of site I, Tyr26 to Tyr7 of site I, Thr28 to Thr9 of site I and Glu31 to Glu12 of sites I; Asp58 corresponds to Asp3 of site II, Asp64 to Asp9 of site II and Glu67 to Glu12 of site II.

**Table 1 biomolecules-12-01703-t001:** Amino acid sequences for the investigated mutants of CaM N-ter domain. Modified residues compared with the wild type sequence are reported in red. Residues of the two metal-binding sites are highlighted in bold. Only the amino acids in each metal-binding loop are numbered, from 1 to 12. The first five peptides are impaired to bind UO_2_^2+^ at site I, while all the other variants are able to bind UO_2_^2+^ at both sites; variants named with a W present a tryptophan residue at position 7 of site II; all variants present a tyrosine residue at position 7 of site I. Variants with site I Thr9P are named with a P. Symbol T_P_ represents a phosphorylated threonine residue.

Peptide		Site I													Site II												
		1	2	3	4	5	6	7	8	9	10	11	12		1	2	3	4	5	6	7	8	9	10	11	12	
CaM Y II	MADQLTDDQISEFKEAFSLF	**A**	**K**	**A**	**G**	**D**	**G**	**Y**	**I**	**T**	**T**	**K**	**E**	LGTVMRSLGQNPTEAELQDMINEV	**D**	**A**	**D**	**G**	**N**	**G**	**T**	**I**	**D**	**F**	**P**	**E**	FLNLMARK
CaM1 Y II	MADQLTDDQISEFKEAFSLF	** A **	**K**	** A **	**G**	**D**	**G**	** Y **	**I**	**T**	** A **	** A **	**E**	LGTVMRSLGQNPTEAELQDMINEV	**D**	**A**	**D**	**G**	**N**	**G**	**T**	**I**	**D**	**F**	**P**	**E**	FLNLMARK
CaM YW II	MADQLTDDQISEFKEAFSLF	** A **	**K**	** A **	**G**	**D**	**G**	** Y **	**I**	**T**	**T**	**K**	**E**	LGTVMRSLGQNPTEAELQDMINEV	**D**	**A**	**D**	**G**	**N**	**G**	** W **	**I**	**D**	**F**	**P**	**E**	FLNLMARK
CaM1 YW II	MADQLTDDQISEFKEAFSLF	** A **	**K**	** A **	**G**	**D**	**G**	** Y **	**I**	**T**	** A **	** A **	**E**	LGTVMRSLGQNPTEAELQDMINEV	**D**	**A**	**D**	**G**	**N**	**G**	** W **	**I**	**D**	**F**	**P**	**E**	FLNLMARK
CaM1 Y II P	MADQLTDDQISEFKEAFSLF	** A **	**K**	** A **	**G**	**D**	**G**	** Y **	**I**	** T_P_ **	** A **	** A **	**E**	LGTVMRSLGQNPTEAELQDMINEV	**D**	**A**	**D**	**G**	**N**	**G**	**T**	**I**	**D**	**F**	**P**	**E**	FLNLMARK
CaM Y I-II	MADQLTDDQISEFKEAFSLF	**D**	**K**	**D**	**G**	**D**	**G**	** Y **	**I**	**T**	**T**	**K**	**E**	LGTVMRSLGQNPTEAELQDMINEV	**D**	**A**	**D**	**G**	**N**	**G**	**T**	**I**	**D**	**F**	**P**	**E**	FLNLMARK
CaM1 Y I-II	MADQLTDDQISEFKEAFSLF	**D**	**K**	**D**	**G**	**D**	**G**	** Y **	**I**	**T**	** A **	** A **	**E**	LGTVMRSLGQNPTEAELQDMINEV	**D**	**A**	**D**	**G**	**N**	**G**	**T**	**I**	**D**	**F**	**P**	**E**	FLNLMARK
CaM YW I-II	MADQLTDDQISEFKEAFSLF	**D**	**K**	**D**	**G**	**D**	**G**	** Y **	**I**	**T**	**T**	**K**	**E**	LGTVMRSLGQNPTEAELQDMINEV	**D**	**A**	**D**	**G**	**N**	**G**	** W **	**I**	**D**	**F**	**P**	**E**	FLNLMARK
CaM1 YW I-II	MADQLTDDQISEFKEAFSLF	**D**	**K**	**D**	**G**	**D**	**G**	** Y **	**I**	**T**	** A **	** A **	**E**	LGTVMRSLGQNPTEAELQDMINEV	**D**	**A**	**D**	**G**	**N**	**G**	** W **	**I**	**D**	**F**	**P**	**E**	FLNLMARK
CaM1 Y I-II P	MADQLTDDQISEFKEAFSLF	**D**	**K**	**D**	**G**	**D**	**G**	** Y **	**I**	** T_P_ **	** A **	** A **	**E**	LGTVMRSLGQNPTEAELQDMINEV	**D**	**A**	**D**	**G**	**N**	**G**	**T**	**I**	**D**	**F**	**P**	**E**	FLNLMARK
CaM1 YW I-II P	MADQLTDDQISEFKEAFSLF	**D**	**K**	**D**	**G**	**D**	**G**	** Y **	**I**	** T_P_ **	** A **	** A **	**E**	LGTVMRSLGQNPTEAELQDMINEV	**D**	**A**	**D**	**G**	**N**	**G**	** W **	**I**	**D**	**F**	**P**	**E**	FLNLMARK

**Table 2 biomolecules-12-01703-t002:** Macroscopic binding constants for reaction of UO_2_^2+^ and Ca^2+^ ions with the investigated mutants of CaM N-terminal domain. I = 0.12 M (MES, KCl); T = 298 K. All reported data are averages of three experimental values.

Peptide	pH		UO_2_^2+^			Ca^2+^	
		K_1_ (M^−1^)	Kd_1_ (nM)	K_2_ (M^−1^)	Kd_2_ (nM)	K_1_ (M^−1^)	K_2_ (M^−1^)
CaM Y II	6	^a^ (3.7 ± 0.5) × 10^6^ ^b^ (4.2 ± 1.0) × 10^6^	270 ± 42 238 ± 74	-		^c^ (1.8 ± 0.3) × 10^4^	-
CaM1 Y II	6	^a^ (3.5 ± 0.7) × 10^6^ ^b^ (3.9 ± 0.5) × 10^6^	286 ± 47 256 ± 39	-		^a^ (2.2 ± 0.2) × 10^4^	-
CaM YW II	6	^a^ (5.1 ± 0.5) × 10^6^ ^b^ (4.9 ± 1.3) × 10^6^	196 ± 21 204 ± 74	-		^c^ (3.1 ± 0.3) × 10^4^	-
CaM1 YW II	6	^a^ (5.9 ± 0.6) × 10^6^ ^b^ (9.0 ± 1.2) × 10^6^	169 ± 20 111 ± 18	-		^a^ (4.7 ± 0.2) × 10^4^	-
CaM Y I	6	^d^ (4.0 ± 0.2) × 10^7^	25 ± 1	^-^		^d^ (2.6 ± 0.1) × 10^4^	^-^
CaM1 Y I	6	^d^ (3.1 ± 0.1) × 10^7^	32 ± 1	^-^		^d^ (4.3 ± 0.4) × 10^4^	^-^
CaM1 Y II P	6 7	^a^ (2.1 ± 0.4) × 10^7^ ^a^ (1.8 ± 0.4) × 10^8^	48 ± 11 5.5 ± 1	^-^		^-^	^-^
CaM Y I-II	6	^a^ (3.5 ± 1.0) × 10^7^ ^b^ (4.4 ± 0.7) × 10^7^	29 ± 6 23 ± 3	^a^ (7.4 ± 0.8) × 10^6^ ^b^ (5.7 ± 1.0) × 10^6^	135 ± 13 175 ± 26	^c^ (5.3 ± 0.3) × 10^4^	^c^ (1.4 ± 0.3) × 10^5^
CaM1 Y I-II	6	^a^ (3.0 ± 0.9) × 10^7^ ^b^ (4.5 ± 0.6) × 10^7^	33 ± 7 22 ± 3	^a^ (7.0 ± 1.4) × 10^6^ ^b^ (6.5 ± 0.7) × 10^6^	143 ± 24 154 ± 15	^a^ (6.0 ± 0.6) × 10^4^	^a^ (1.2 ± 0.3) × 10^5^
CaM YW I-II	6	^a^ (7.4 ± 1.5) × 10^7^ ^b^ (6.3 ± 0.5) × 10^7^	14 ± 3 16 ± 2	^a^ (9.7 ± 1.0) × 10^6^ ^b^ (7.8 ± 0.3) × 10^6^	103 ± 10 128 ± 5	^c^ (8.1 ± 0.3) × 10^4^	^c^ (3.1 ± 0.3) × 10^5^
CaM1 YW I-II	6	^a^ (7.8 ± 0.3) × 10^7^ ^b^ (7.0 ± 0.6) × 10^7^	13 ± 1 14 ± 2	^a^ (9.4 ± 1.5) × 10^6^ ^b^ (8.5 ± 0.5) × 10^6^	106 ± 15 118 ± 7	^a^ (7.7 ± 0.4) × 10^4^	^a^ (2.8 ± 0.2) × 10^5^
CaM1 Y I-II P	6 7	^a^ (3.6 ± 1.0) × 10^8^ ^a^ (1.0 ± 0.5) × 10^11^	2.8 ± 0.6 0.010 ± 0.008	^a^ (7.8 ± 0.7) × 10^7^ ^a^ (7.1 ± 1.0) × 10^8^	13 ± 2 1.4 ± 0.2	^a^ (7.1 ± 0.2) × 10^4^ ^a^ (8.9 ± 0.4) × 10^4^	^a^ (1.0 ± 0.7) × 10^5^ ^a^ (1.1 ± 0.8) × 10^5^
CaM1 YW I-II P	6 7	^a^ (3.8 ± 1.0) × 10^8^ ^a^ (6.0 ± 0.9) × 10^11^	2.6 ± 0.6 0.0017 ± 0.0003	^a^ (1.3 ± 0.5) × 10^8^ ^a^ (5.1 ± 0.6) × 10^9^	7.7 ± 2.2 0.196 ± 0.021	^a^ (1.0 ± 0.3) × 10^5^ ^a^ (9.0 ± 0.5) × 10^4^	^a^ (2.2 ± 0.1) × 10^5^ ^a^ (1.1 ± 0.2) × 10^5^

^a^ Data from spectrofluorimetric titrations. ^b^ Data from calorimetric titrations. ^c^ Data from Beccia et al. (2015) [28]. ^d^ Data from Pardoux et al. (2012) [16].

**Table 3 biomolecules-12-01703-t003:** Thermodynamic macroscopic parameters for UO_2_^2+^ binding to the investigated mutants of CaM N-terminal domain. pH = 6, I = 0.12 M (MES, KCl); T = 298 K.

Peptide	ΔH_I_ (kJ mol^−1^)	TΔS_I_ (kJ mol^−1^)	ΔH_II_ (kJ mol^−1^)	TΔS_II_ (kJ mol^−1^)	ΔH_c_(kJ mol^−1^)	TΔS_c_ (kJ mol^−1^)
CaM Y II	-	-	−2.0 ± 0.9	35.8 ± 0.9		
CaM1 Y II	-	-	−3.5 ± 1.0	34.1 ± 1.0		
CaM YW II	-	-	−8.9 ± 0.7	29.3 ± 0.8		
CaM1 YW II	-	-	−15.0 ± 2.5	24.7 ± 2.5		
CaM Y I-II	−40.2 ± 0.5	3.2 ± 0.5	−2.0 ± 0.9	35.8 ± 0.9	25.0 ± 0.1	27.0 ± 0.1
CaM1 Y I-II	−26.1 ± 0.3	8.6 ± 0.3	−3.5 ± 1.0	34.1 ± 1.0	22.0 ± 0.2	5.8 ± 0.2
CaM YW I-II	−34.8 ± 0.1	18.2 ± 0.1	−8.9 ± 0.7	29.3 ± 0.8	3.8 ± 0.1	10.7 ± 0.1
CaM1 YW I-II	−26.1 ± 0.3	44.5 ± 0.3	−15.0 ± 5.0	24.7 ± 2.5	8.9 ± 0.3	1.4 ± 0.3

**Table 4 biomolecules-12-01703-t004:** Cooperativity parameters and microscopic binding constants for reaction of UO_2_^2+^ ion with the two-site mutants of CaM N-terminal domain, evaluated using spectrofluorimetric experimental data. I = 0.12 M (MES, KCl); T = 298 K.

Peptide	pH	ΔΔG (kJ mol^−1^)	K_I_ (M^−1^)	K_II_ (M^−1^)	K_I,II_ (M^−1^)	K_II,I_ (M^−1^)
CaM Y I-II	6	−2.0 ± 0.4	(3.1 ± 0.3) × 10^7^	(3.7 ± 0.5) × 10^6^	(7.1 ± 0.2) × 10^7^	(8.3 ± 0.3) × 10^6^
CaM1 Y I-II	6	−2.0 ± 1.0	(2.7 ± 0.3) × 10^7^	(3.5 ± 0.7) × 10^6^	(6.0 ± 0.2) × 10^7^	(7.9 ± 0.9) × 10^6^
CaM YW I-II	6	−1.8 ± 0.3	(6.8 ± 0.2) × 10^7^	(5.1 ± 0.5) × 10^6^	(1.4 ± 0.1) × 10^8^	(1.0 ± 0.2) × 10^7^
CaM1 YW I-II	6	−1.3 ± 0.2	(7.2 ± 0.3) × 10^7^	(5.9 ± 0.6) × 10^6^	(1.2 ± 0.1) × 10^8^	(1.0 ± 0.2) × 10^7^
CaM1 Y I-II P	6	−3.4 ± 0.4	(3.4 ± 0.3) × 10^8^	(2.1 ± 0.4) × 10^7^	(1.3 ± 0.2) × 10^9^	(8.3 ± 0.3) × 10^7^
	7	−3.4 ± 0.2	(1.0 ± 0.5) × 10^11^	(1.8 ± 0.4) × 10^8^	(3.9 ± 0.5) × 10^11^	(7.1 ± 0.1) × 10^8^
CaM1 YW I-II P	6	−4.7 ± 0.5	(3.6 ± 0.3) × 10^8^	(2.1 ± 0.4) × 10^7^	(2.4 ± 0.2) × 10^9^	(1.4 ± 0.5) × 10^8^
	7	−8.3 ± 0.3	(6.0 ± 0.9) × 10^11^	(1.8 ± 0.4) × 10^8^	(1.7 ± 0.5) × 10^13^	(5.1 ± 0.2) × 10^9^

**Table 5 biomolecules-12-01703-t005:** Analysis of the coordination sphere of the two uranyl molecules in CaM1 Y I-II P from the structural model obtained by molecular dynamics simulation. Mean of distances (in Å) calculated from the 1500 last steps of the 15 ns simulation.

**Site I**	**U_1_-OAsp3**	**U_1_-OAsp5**	**U_1_-OTyr7**	**U_1_-OGlu12**
Mean Distance (Å)	2.35	2.33	2.46	2.35 2.35
**Site II**	**U_2_-OAsp3-**	**U_2_-O H_2_O**	**U_2_-OAsp9**	**U_2_-OGlu12**
Mean Distance (Å)	2.32	2.36	2.39	2.37 2.37

## Data Availability

Data are available upon request.

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
