# Peer review of "Inter-Site Cooperativity of Calmodulin N-Terminal Domain and Phosphorylation Synergistically Improve the Affinity and Selectivity for Uranyl"

_biomolecules, 2022, doi:10.3390/biom12111703_

Round 1

Reviewer 1 Report

In general, the topic of this work is really interesting. Quantitative studies on the affinity of uranyl or other actinides or radionuclides to biomolecules are crucial to understand the binding mechanisms, toxicological effects and to develop effective decontamination and decorporation strategies. The study was performed very carefully, the results are comprehensible and the manuscript was written very detailed and clear. However, some comments could be considered before publication.

-          Results (3.2, line 378): It is not clear to the reader how the 1:1 complex formation (the stoichiometry) was identified from the titration curves.

-          Generally: What does the higher affinity to uranyl compared to calcium mean in natural systems: (i) which uranyl concentrations could be reached in e.g. accidental cases? (ii) At which concentration ratios could uranyl displace calcium? Did you perform such kind of experiments?

Author Response

Reviewer 1

In general, the topic of this work is really interesting. Quantitative studies on the affinity of uranyl or other actinides or radionuclides to biomolecules are crucial to understand the binding mechanisms, toxicological effects and to develop effective decontamination and decorporation strategies. The study was performed very carefully, the results are comprehensible and the manuscript was written very detailed and clear. However, some comments could be considered before publication.

- Results (3.2, line 378): It is not clear to the reader how the 1:1 complex formation (the stoichiometry) was identified from the titration curves.

The conclusion to the formation of a 1 :1 complex resulted from the fact that the experimental data were satisfactorily fitted using binding isotherms corresponding to the formation of 1:1 complexes.  There was no need to use more complex 1:2 or 2:1 models. We modified the text to make this more clear line 381.

       - Generally: What does the higher affinity to uranyl compared to calcium mean in natural systems: (i) which uranyl concentrations could be reached in e.g. accidental cases? (ii) At which concentration ratios could uranyl displace calcium? Did you perform such kind of experiments?

The affinities of calmodulin’s site I and site II are 200 (site II) to 1000 (site I) times higher for uranyl than for calcium. In natural systems, the calmodulin will bind calcium only if the intracellular calcium concentration increases in the cytoplasm, to reach concentrations of calcium in the micromolar to tenth of micromolar range. In these conditions, nanomolar concentrations of uranyl (i.e., a calcium / uranyl concentration ratio of about one thousand) should already result in partial binding of uranyl to the metal binding sites of calmodulin. It is difficult to anticipate local concentrations of uranium that could be reached in accidental cases. We did not perform such kind of competition experiments with the calmodulin variants used here. This could be done in a future study.

Reviewer 2 Report

The submitted manuscript provides a well studied in depth understanding of the interaction between UO2 and calmodium that contributes to the understanding of the potential biological effect of uranium toxicity.

I'd recommend publication, very few minor points should be addressed in the proof-reading:

1- the authors use two formats for numbers, "6.0x106" (e.g. caption figure 2) and "6.0.106" (e.g. line 582). I'd recommend only one format is used for consistency.

2- the authors mention amino acids such as Asp3, Asp5, Try7 (line 657). These numbers seem to refer to the sequence at each site (table 1). This means that there are two Gly4 amino acids, one on each site. Also, Figure 6 gives the AA with their full sequence numbering. Hence Asp3 of site I (Table 1) is probably actually Asp24 of the whole sequence in Figure 6. The main numbering format (that of table 1) is not easy to follow, especially if we consider these two sites are not duplicates of each other. The authors should consider using the full sequence numbering throughout, i.e. the one used in figure 6.

Author Response

Reviewer 2

The submitted manuscript provides a well studied in depth understanding of the interaction between UO2 and calmodium that contributes to the understanding of the potential biological effect of uranium toxicity.

I'd recommend publication, very few minor points should be addressed in the proof-reading:

1- The authors use two formats for numbers, "6.0x106" (e.g. caption figure 2) and "6.0.106" (e.g. line 582). I'd recommend only one format is used for consistency.

Changes were done with the format "6.0.106".

2- The authors mention amino acids such as Asp3, Asp5, Try7 (line 657). These numbers seem to refer to the sequence at each site (table 1). This means that there are two Gly4 amino acids, one on each site. Also, Figure 6 gives the AA with their full sequence numbering. Hence Asp3 of site I (Table 1) is probably actually Asp24 of the whole sequence in Figure 6. The main numbering format (that of table 1) is not easy to follow, especially if we consider these two sites are not duplicates of each other. The authors should consider using the full sequence numbering throughout, i.e. the one used in figure 6.

There is no easy way to number the amino acids in the text. In the paper, we have chosen to number the amino acids of each of the two metal binding sites from 1 to 12, for simplification, considering that they are the amino acids that play a role in metal binding and that their nature and position in the 1 to 12 sequence is also important. We always mention if we refer to site I or site II in the text. In figure 6, the two sites are represented and we thought that it would have been confusing if we had kept the numbering with two amino acids #1, two amino acids #2, and so on… This is why we preferred to use the global numbering of the protein. We have added a sentence in the legend of Figure 6 which details the correspondence between the two numberings, to make it clearer for the reader.

Reviewer 3 Report

In this paper, the authors used spectrofluorimetry and ITC to evaluate thermodynamic parameters for the interaction of uranyl with the two calcium-binding sites of calmodulin, in a series of engineered mutants. They showed in particular that the affinity and the selectivity for uranyl against calcium can be increased by phosphorylation of a Thr residue at site I. The work is well designed, and the results are appropriately presented and discussed. Therefore, I can recommend publication in Biomolecules. I have just a few comments/typos caught:

- Page 6, line 233: “for whom” should be “for which”

- Page 10, line 398: standard deviation value is missing for KdI (it should be 1)

- Page 15, line 533 and page 16, line 556: “circa” may be better in italic (line 630)

- Page 19, line 708: “while any” should be “while no” (if this is what the authors mean)

- Figure 6: the numbering of residues is different from that used in the text, making difficult for the reader to follow the discussion of the figure. Furthermore, I would include Tyr26 in the B panel to allow a complete comparison, even if it has no atom closer than 2.5 A to the uranyls.

Author Response

Reviewer 3

In this paper, the authors used spectrofluorimetry and ITC to evaluate thermodynamic parameters for the interaction of uranyl with the two calcium-binding sites of calmodulin, in a series of engineered mutants. They showed in particular that the affinity and the selectivity for uranyl against calcium can be increased by phosphorylation of a Thr residue at site I. The work is well designed, and the results are appropriately presented and discussed. Therefore, I can recommend publication in Biomolecules. I have just a few comments/typos caught:

- Page 6, line 233: “for whom” should be “for which” Done

- Page 10, line 398: standard deviation value is missing for KdI (it should be 1) Done

- Page 15, line 533 and page 16, line 556: “circa” may be better in italic (line 630) Done

- Page 19, line 708: “while any” should be “while no” (if this is what the authors mean) Done

- Figure 6: the numbering of residues is different from that used in the text, making difficult for the reader to follow the discussion of the figure.

There is no easy way to number the amino acids in the text. In the paper, we have chosen to number the amino acids of each of the two metal binding sites from 1 to 12, for simplification, considering that they are the amino acids that play a role in metal binding and that their nature and position in the 1 to 12 sequence is also important. We always mention if we refer to site I or site II in the text. This has been added in the text lines 363-364. In figure 6, the two sites are represented and we thought that it would have been confusing if we had kept the numbering with two amino acids #1, two amino acids #2, and so on… This is why we preferred to use the global numbering of the protein. We have added a sentence in the legend of Figure 6 which details the correspondence between the two numberings, to make it clearer for the reader.

Furthermore, I would include Tyr26 in the B panel to allow a complete comparison, even if it has no atom closer than 2.5 A to the uranyls.

We drew a new figure (see below) adding Tyr26, as required. However, Tyr26 masks Asp64 and it is much less easy to see the two complexation sites. In the figure, Tyr26 side-chain could seem to participate to uranyl coordination at site II. Therefore, we prefer the figure 6 that we had originally proposed.

(see the figure in the attached file please)
